# Regional Accuracy Assessment of 30-Meter GLC_FCS30, GlobeLand30, and CLCD Products: A Case Study in Xinjiang Area

Jingpeng Liu [1], Yu Ren [2] and Xidong Chen [1,3,*]

1 College of Surveying and Geo-Informatics, North China University of Water Resources and Electric Power, Zhengzhou 450046, China; 201904306@stu.ncwu.edu.cn
2 College of Geography and Environmental Sciences, Northwest Normal University, Lanzhou 730070, China; 2021222961@nwnu.edu.cn
3 Future Urbanity & Sustainable Environment (FUSE) Lab, Division of Landscape Architecture, Department of Architecture, Faculty of Architecture, The University of Hong Kong, Hong Kong SAR 999007, China
* Correspondence: xdchenrs@hku.hk

**Abstract:** With the development of remote sensing technology, a number of fine-resolution (30-m) global/national land cover (LC) products have been developed. However, accuracy assessments for the developed LC products are commonly conducted at global and national scales. Due to the limited availability of representative validation observations and reference data, knowledge relating to the accuracy and applicability of existing LC products on a regional scale is limited. Since Xinjiang, China, exhibits diverse surface cover and fragmented urban landscapes, existing LC products generally have high classification uncertainty in this region. This makes Xinjiang suitable for assessing the accuracy and consistency of exiting fine-resolution land cover products. In order to improve knowledge of the accuracy of existing fine-resolution LC products at the regional scale, Xinjiang province was selected as the case area. First, we employed an equal-area stratified random sampling approach with climate, population density, and landscape heterogeneity information as constraints, along with the hexagonal discrete global grid system (HDGGS) as basic sampling grids to develop a high-density land cover validation dataset for Xinjiang (HDLV-XJ) in 2020. This is the first publicly available regionally high-density validation dataset that can support analysis at a regional scale, comprising a total of 20,932 validation samples. Then, based on the generated HDLV-XJ dataset, the accuracies and consistency among three widely used 30-m LC products, GLC_FCS30, GlobeLand30, and CLCD, were quantitatively evaluated. The results indicated that the CLC_FCS30 exhibited the highest overall accuracy (88.10%) in Xinjiang, followed by GlobeLand30 (with an overall accuracy of 83.58%) and CLCD (81.57%). Moreover, through a comprehensive analysis of the relationship between different environmental conditions and land cover product performance, we found that GlobeLand30 performed best in regions with high landscape fragmentation, while GLC_FCS30 stood out as the most outstanding product in areas with uneven proportions of land cover types. Our study provides a novel insight into the suitability of these three widely-used LC products under various environmental conditions. The findings and dataset can provide valuable insights for the application of existing LC products in different environment conditions, offering insights into their accuracies and limitations.

**Keywords:** land cover; remote sensing; validation dataset; accuracy assessment; consistency analysis; stratified random sampling

## 1. Introduction

Land cover (LC) monitoring plays a pivotal role in ecological environment governance, agricultural and forestry management, urban and rural planning, and ecological

conservation. The rapid development of computer technology and the continuous sharing of medium- to fine-resolution remote sensing data, such as from Landsat and Sentinel, have ushered the world into an era of fine-resolution remote sensing [1,2]. When conducting large-scale macro-scale land cover research, a resolution of 30 m is considered optimal [3,4]. Moreover, this resolution has an advantageous data size, enabling more efficient data processing and storage.

In recent years, many 30-m fine-resolution LC products have been produced. For example, Chen et al. (2015) developed the well-known GlobeLand30 using a pixel-object-knowledge-based (POK) classification strategy based on multi-temporal Landsat and HJ-1A/B imagery [3], Zhang et al. (2021) generated a global-scale 30-m Global Land Cover with Fine Classification System (GLC_FCS30) product based on the spectral library of land cover and time-series Landsat data [1], and Yang et al. (2021) utilized Landsat data and the random forest classifier to generate the annual China Land Cover Dataset (CLCD) based on the Google Earth Engine (GEE) [5]. Similarly, Gong et al. (2013) produced the Finer Resolution Observation and Monitoring of Global Land Cover (FROM_GLC) product using the Landsat thematic mapper (TM) and enhanced thematic mapper Plus (ETM+) data [6]. The United States Geological Survey (USGS) collaborated with multiple federal agencies to produce the 30-m National Land Cover Database (NLCD), which covers the entire United States [7]. These large-scale fine-resolution LC products provide adequate information to observe the fundamental characteristics and trends of land cover.

However, the accuracies reported for the abovementioned large-scale LC products are generally towards global and national scales (Table 1). For instance, the accuracy of GlobeLand30 was evaluated using 38,644 globally distributed validation samples [3]. Similarly, the accuracy of GLC_FCS30 was assessed using 44,043 samples that were distributed across the globe [1]. For the assessment of CLCD accuracy, a total of 5463 validation points within China were used [5]. Nevertheless, due to the differences in scene complexity and data quality across regions, the accuracy of land cover products inevitably varies from region to region [3,8–10]. In some complex areas, the accuracy of LC products is even much lower than the overall accuracy of the report [11]. For example, the overall accuracy of GlobeLand30 is reported to be 83.5% worldwide. However, its accuracy significantly drops to approximately 50% in Africa [12]. Similarly, in the Central Asian region, the accuracy of GlobeLand30 was found to be only about 46% [13]. Cui et al. (2023) collected thirteen sets of global or national-scale land cover datasets. Through the visual interpretation of high-resolution images, ground "truth" samples were collected to evaluate the data accuracy across Northeast China [14]. Accordingly, the quantitative evaluation of the accuracy and consistency of existing 30-m land cover products at the regional scale is of great significance for the scientific use of these products.

**Table 1.** Summary of existing accuracy evaluation studies of 30-m LC products.

| Validated LC Products | Validation Area | Sample Quantity | Literature |
|---|---|---|---|
| GLC_FCS30 | Globe | 44,043 | Zhang et al. (2021) [1] |
| CLCD | China | 5463 | Yang et al. (2021) [5] |
| GlobeLand30 | Globe | 38,644 | Chen et al. (2015) [3] |
| FROM-GLC30 | Globe | 38,664 | Zhao et al. (2014) [15] |
| GlobeLand30, FROM-GLC30, and GLC_FCS30 | Globe | 79,112 | Zhao et al. (2023) [16] |

The validation of land cover products is typically conducted by randomly selecting validation points and visually interpreting the labels of each point using high-resolution imagery for comparison with the product [17]. Therefore, in order to accurately evaluate the accuracy of land cover products in a specific region, a scientific high-density validation dataset is a crucial prerequisite [10,15,18–20]. The construction of a high-density verification dataset in a scientific manner, with sufficient quantities of representative samples

covering different categories in each region, is a crucial issue. The research conducted by Sahr (2011) demonstrated that dividing the study area into equal-sized grids and assigning validation points at the grid level ensures coverage of sample points in all regions [21]. Zhao et al. (2023) demonstrated that considering multiple factors such as climate and population when allocating sampling points and employing a stratified random sampling approach can result in a more reasonable distribution of validation points [16]. To ensure an adequate number of samples in complex areas and appropriate placement of sampling points in homogeneous regions, this study employs a multi-indicator equal-area stratified random sampling method [15]. This approach not only increases the sample size of rare land cover categories but also effectively reduces the standard deviation of accuracy assessment [16]. Additionally, the use of equal-area stratified random sampling ensures the balance of different categories during the sample selection process, thereby enhancing the representativeness and reliability of the assessment results.

To this end, we first employed an equal-area stratified random sampling approach with climate, population density, and landscape heterogeneity information as constraints, along with the hexagonal discrete global grid system (HDGGS) as basic sampling grids to develop a High-Density Land cover Validation dataset in Xinjiang (HDLV-XJ). Then, the consistency and accuracy of three widely used 30-m resolution LC products (GLC_FCS30, CLCD, GlobeLand30) in 2020 were analyzed based on the developed validation dataset. Additionally, the suitability of these three LC products under different environment conditions was for the first time analyzed and revealed. The results of this study can provide an important reference value for the accuracy assessment of existing LC products on regional scales and for the application of LC products in specific scenarios. Additionally, our HDLV-XJ dataset can provide valuable data support for regional-scale research.

## 2. Study Area and Data

### 2.1. Study Area

The study area chosen for this research was the Xinjiang Uyghur Autonomous Region, situated between 73°40′ and 96°18′ east longitude and 34°25′ and 48°10′ north latitude (Figure 1). As the largest provincial-level administrative region in China, Xinjiang encompasses an extensive land area, accounting for over one-sixth of the country's total land area, with a precise measurement of 1,664,897 km$^2$ [22]. Located in the Northwestern part of China, Xinjiang lies in the hinterland of the Eurasian continent and is considered one of the provinces farthest from the ocean within China. Xinjiang has a typical temperate continental climate that has only a few and unevenly distributed precipitation events [23]. The region comprises diverse land cover types, including primarily deserts, grasslands, and croplands. The distribution of vegetation is predominantly observed in mountainous and oases [24].

### 2.2. The 30-Meter Global Land Cover Products

Three widely used LC products were selected for evaluation: GlobeLand30, GLC_FCS30, and CLCD (Figure 2). Among them, GLC_FCS30 is a 30-m global land cover product developed using time-series Landsat data. The product covers the time span from 1985 to 2020, with update cycles occurring every 5 years. It incorporates 16 primary LCCS land cover types and 14 secondary land cover types (Table 2). The overall global accuracy of the GLC_FCS30 is reported as 82.5% [1]. CLCD is a 30-m-resolution annual land cover dataset for China. This product is derived from the utilization of Landsat data in conjunction with a random forest classifier. The dataset includes yearly land cover information for China from 1985 to 2019. Its classification system includes nine land cover types (Table 2). The overall accuracy of the CLCD reaches 79.31% [5]. GlobeLand30 is a high-precision global land cover product with a spatial resolution of 30 m. It was updated with three sets of data, in 2000, 2010, and 2020. The development of GlobeLand30 was based on the "Pixel-Object-Knowledge" (POK) method. Its classification system includes 10 land cover types (Table 2), and the overall accuracy is 83.5% [3]. By utilizing the vector boundaries of Xinjiang, the original LC products were clipped and reclassified to standardize their classification system, as shown in Figure 2.

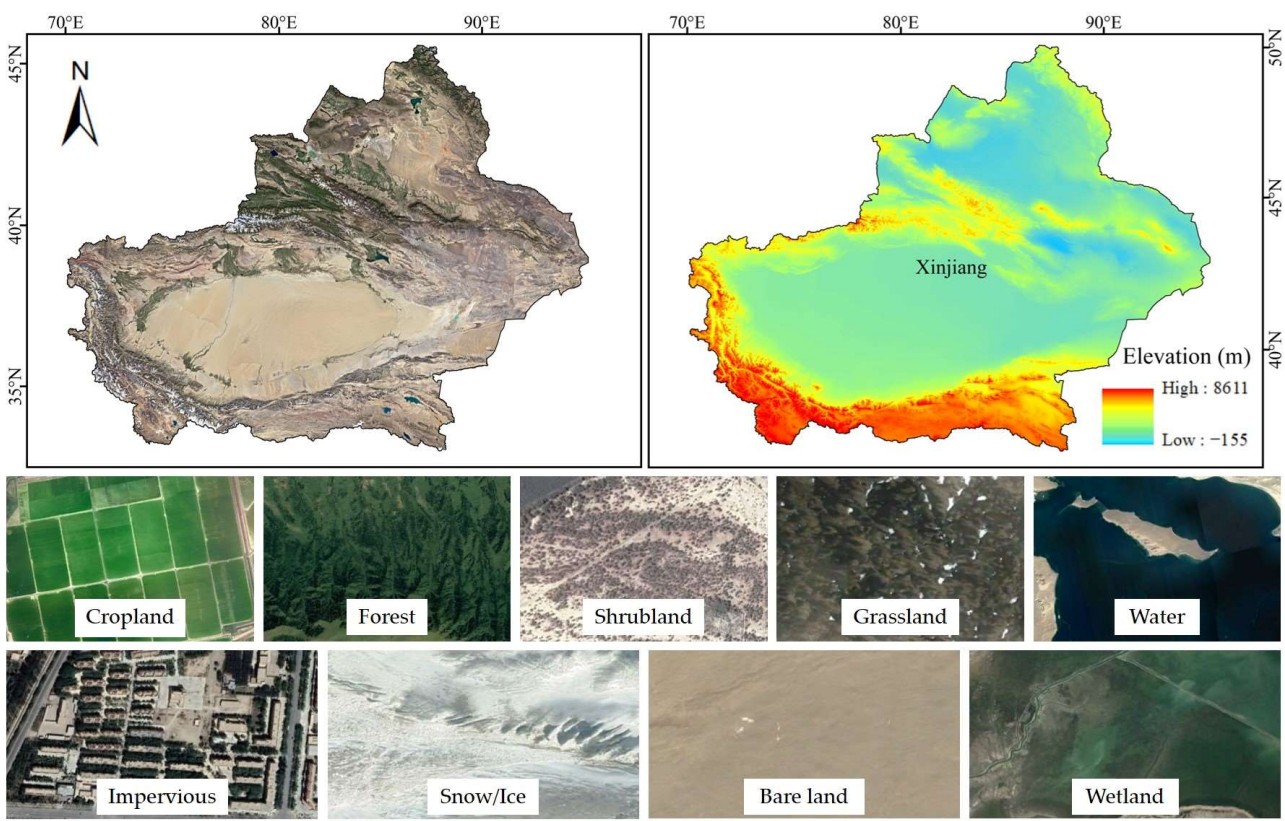

**Figure 1.** Study area and the land cover type along with the corresponding examples of their appearance on Google Earth: (source: Google Earth).

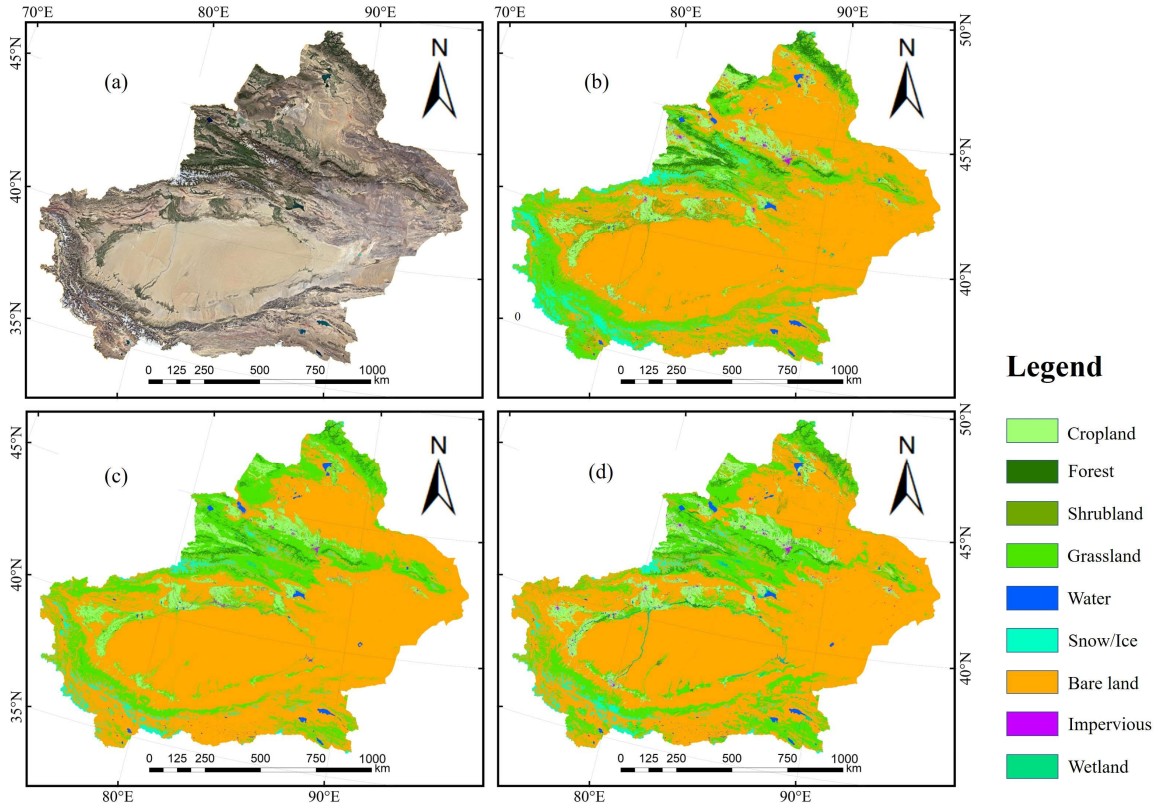

**Figure 2.** The 30-m global land cover products in harmonized classification systems: (**a**) remote sensing image of Xinjiang, (**b**) GLC_FCS30, (**c**) CLCD, and (**d**) GlobeLand30.

**Table 2.** The classification systems of the three LC products. The classification system for GLC_FCS30, CLCD, and GlobeLand30 were sourced from Zhang et al. (2021) [1], Yang et al. (2021) [5], and Chen et al. (2015) [3], respectively. Note: the land cover types we used in this study are in bold.

| CLCD | Id | GlobeLand30 | Id | GLC_FCS30 | Id |
|---|---|---|---|---|---|
| Cropland | 1 | Cultivated land | 10 | Rain-fed cropland | 10 |
| | | | | Herbaceous cover | 11 |
| | | | | Tree or shrub cover (orchard) | 12 |
| | | | | Irrigated cropland | 20 |
| Forest | 2 | Forest | 20 | Evergreen broadleaved forest | 50 |
| | | | | Deciduous broadleaved forest | 60 |
| | | | | Closed deciduous broadleaved forest | 61 |
| | | | | Open deciduous broadleaved forest | 62 |
| | | | | Evergreen needleleaved forest | 70 |
| | | | | Closed evergreen needleleaved forest | 71 |
| | | | | Open evergreen needleleaved forest | 72 |
| | | | | Deciduous needleleaved forest | 80 |
| | | | | Closed deciduous needleleaved forest | 81 |
| | | | | Open deciduous needleleaved forest | 82 |
| | | | | Mixed-leaf forest | 90 |
| Shrub | 3 | Shrubland | 40 | Shrubland | 120 |
| | | | | Evergreen shrubland | 121 |
| | | | | Deciduous shrubland | 122 |
| Grassland | 4 | Grassland | 30 | Grassland | 130 |
| Wetland | 9 | Wetland | 50 | Wetlands | 180 |
| Impervious | 8 | Artificial surfaces | 80 | Impervious surfaces | 190 |
| Bare land | 7 | Bare land | 90 | Lichens and mosses | 140 |
| | | | | Sparse vegetation | 150 |
| | | | | Sparse shrubland | 152 |
| | | | | Sparse herbaceous cover | 153 |
| | | | | Bare areas | 200 |
| | | | | Consolidated bare areas | 201 |
| | | | | Unconsolidated bare areas | 202 |
| Water | 5 | Water bodies | 60 | Water body | 210 |
| Snow/Ice | 6 | Permanent snow and ice | 100 | Permanent ice and snow | 220 |
| | | Tundra | 70 | | |

*2.3. Climate Zone and Population Datasets*

In order to scientifically generate sufficient validation samples in different regions and to avoid spatial imbalance in sample distribution, climate zone and population datasets were also collected. The climate zone data used in this study were from the Köppen–Geiger climate spatial distribution dataset, which was derived based on temperature and precipitation observations from global climate models (http://koeppen-geiger.vu-wien.ac.at/shifts.htm (accessed on 16 June 2023)) [25]. The climate zones are defined as follows: primary climate groups include equatorial (A), arid (B), warm temperate (C), snow (D), and polar (E). Precipitation categories include desert (W), steppe (S), fully humid (f), summer dry (s), winter dry (w), and monsoonal (m). Temperature categories include hot arid (h), cold arid (k), hot summer (a), warm summer (b), cool summer (c), extremely continental (d), polar frost (F), and polar tundra (T). The Köppen–Geiger climate classification spatial distribution data are shown in Figure 3. We clipped the Köppen-Geiger climate spatial distribution dataset using the vector boundary data of Xinjiang (Figure 3b). Xinjiang includes 10 different climate types, namely BSk, BWk, Dfa, Dfb, Dfc, Dsa, Dsb, Dsc, Dwc, and ET. Due to the limited spatial extent and similar characteristics of certain climate classifications (Dsa, Dsb, Dfa) in Xinjiang, this study employed the merging method utilized by Olofsson et al. (2012) for collecting validation data on global land cover products [25–27]. Classes were manually merged into different classes. For example, Dsa: "Continental,

dry and hot summer", Dsb: "Continental, dry and warm summer", and BSk: "Arid, desert, cold" were merged into one desert class in Xinjiang, and Dsa: "Continental, dry summer, hot summer" and Dsa: "Continental, dry summer, warm summer" were merged with the continental forest class. Table 3 summarizes the initial 10 classes into the final 5 climate classifications.

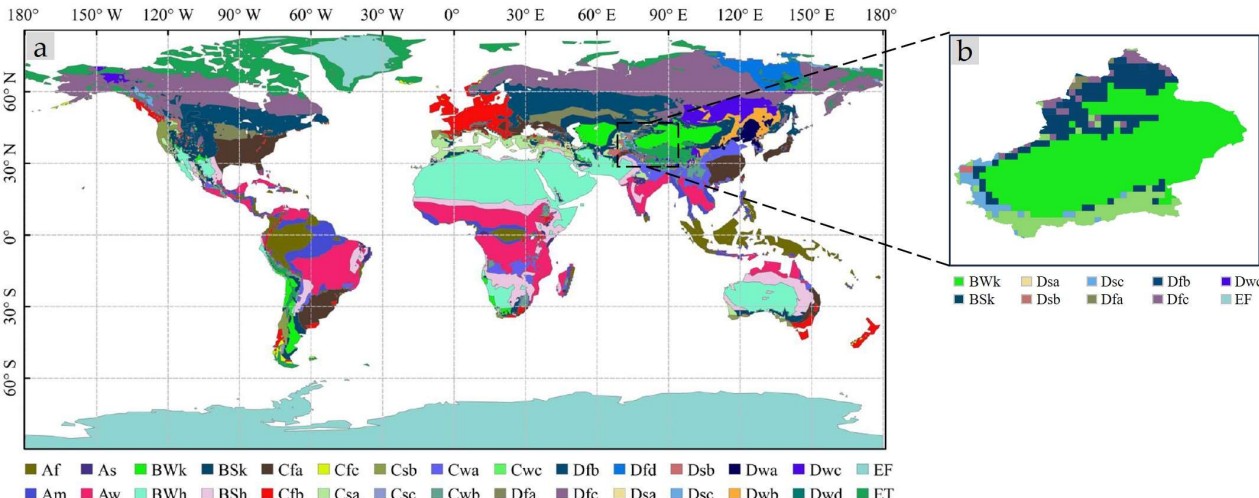

**Figure 3.** The Köppen-Geiger climate classification spatial distribution data for the world (**a**) and Xinjiang (**b**).

**Table 3.** Edited Köppen–Geiger climate classification in Xinjiang.

| New Code | Climate | Climate Code |
|----------|---------|--------------|
| 1 | Desert | BWk |
| 2 | Grassland | BSk + Dsa + Dsb |
| 3 | Continental forest | Dfa + Dfb |
| 4 | Boreal forest | Dsc + Dwc + Dfc |
| 5 | Frost | EF |

The Global Human Settlement Layer (GHSL) dataset provides spatial information on global population distribution and residential areas. It offers four levels of global geospatial population data with resolutions of 100 m, 1 km, and more, covering the period from 1975 to 2030 with a 5-year interval (https://ghsl.jrc.ec.europa.eu/download.php?ds=pop (accessed on 16 June 2023)). Due to the availability of the latest global geospatial population data, GHSL-POP2020 was used as a source for stratified population data in this study. Based on the population density data in GHS-POP2020, the population data of the Xinjiang were reclassified using a threshold of five people per $km^2$ [28], resulting in the creation of the population and un-population layers.

### 2.4. The Existing Land Cover Validation Dataset

Two published global land cover validation datasets were collected in our study to cross-compare with our developed HDLV-XJ and to further illustrate the accuracy of our validation dataset. The first was the global validation dataset (SRS_Val) released by Zhao et al. (2023) [16]. The SRS_Val adopted a stratified random sampling method to distribute 79,112 validation samples globally in 2020, of which 659 validation points were distributed in Xinjiang. The second were the global land cover validation samples (GLV_2015) produced by Zhang et al. (2019) in 2015 [1]. The GLV_2015 was generated through integrating the GLCNMO2008 training dataset, VIIRS reference dataset, STEP reference dataset, global cropland reference data, and high-resolution imagery in Google Earth. This dataset contains a total of 403 sample points in Xinjiang. Both SRS_Val and GLV_2015 adopted a standardized classification system derived from UN-LCCS, and

GLV_2015 is only distributed in six categories in Xinjiang. The quantities and spatial distributions of the validation points in these two datasets are illustrated in Figure 4a,b. The validation samples of SRS_val are scattered throughout Xinjiang. The most prevalent land cover in SRS_Val is bare area (with 296 samples), followed by grassland (with 143 samples) and sparse vegetation (with 83 samples). Conversely, the sample points of GLV_2015 in the Xinjiang region are predominantly concentrated in the south. Furthermore, bare land presented the highest counts in the GLV_2015 (with 222 samples), followed by permanent ice and snow (with 96 samples) and sparse vegetation (with 56 samples). The land types in the two datasets were modified according to the classification system used in this study (Table 2). The number of samples corresponding to each class after adjustment is shown in Figure 4c.

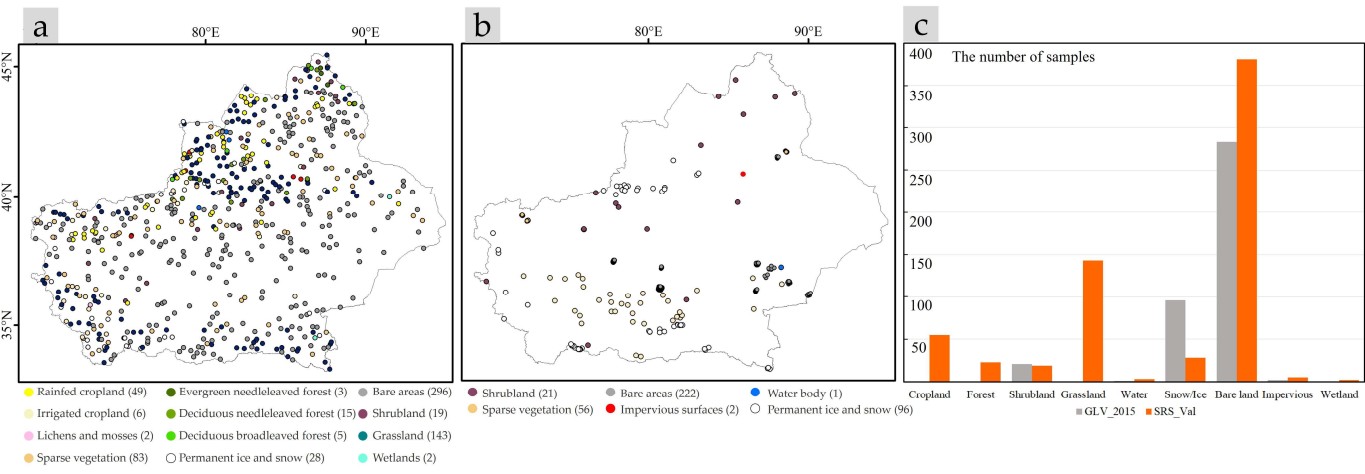

**Figure 4.** The quantities and spatial distributions of (**a**) SRS_Val and (**b**) GLV_2015, and (**c**) the number of samples corresponding to each class after LC classification system adjustment. Note: each number following the class represents the sample quantity for that category.

## 3. Methods

### 3.1. Harmonization of the Land Cover Classification Systems

This study encounters the challenge of dealing with the different classification systems employed by the LC products GLC_FCS30, CLCD, and GlobeLand30. When computing the consistency of LC products, the differences in classification systems used in LC products can affect the consistency and accuracy of those different products [8–10]. The classification systems of CLCD and GlobeLand30 solely encompass land cover's first-level classes, while GLC_FCS30 incorporates three-level classes; therefore, a classification system based on those of CLCD and GlobeLand30 was adopted (Table 2). It should be noted that the tundra in GlobeLand30 was not included in our classification system because this land cover class is almost nonexistent in the Xinjiang region. Meanwhile, the classification system of GLC_FCS30 was merged and adjusted to the categories used in this study (Table 2).

### 3.2. Construction of the Land Cover Validation Dataset in Xinjiang

The flowchart is illustrated in Figure 5. First, we performed a spatial overlay operation on the Köppen–Geiger climate spatial distribution data and GHSL-POP2020 data to obtain a climate–population density constraint layer. Furthermore, the Shannon diversity index (*SHDI*) was calculated to derive the landscape heterogeneity, and the hexagonal discrete global grid system (HDGGS) was used to generate basic sample-allocated hexagonal grids. Then, a spatial join operation was utilized to combine the above constraint layers to determine the number of samples distributed in each hexagonal grid. A total of 22,000 sample points were allocated within each grid based on the derived constraint layers. Finally, the category of each validation sample point was annotated based on visual interpretation to generate the HDLV-XJ.

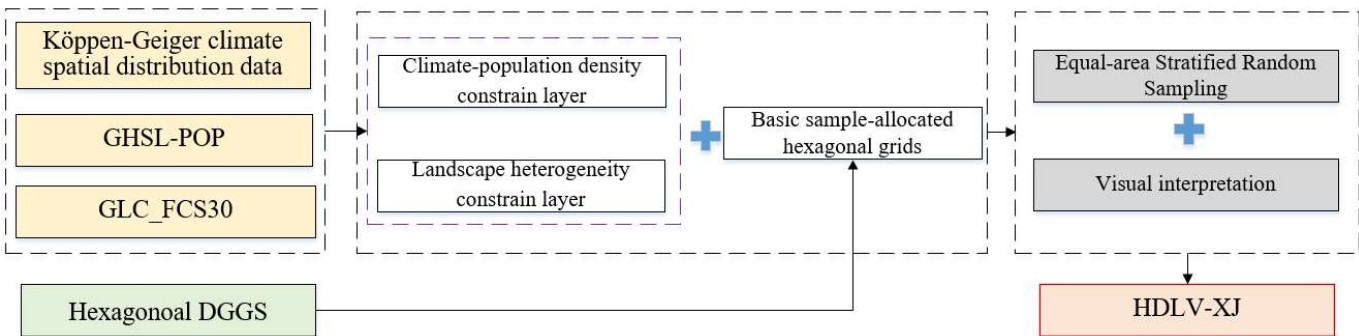

**Figure 5.** Flowchart for constructing the HDLV-XJ.

### 3.2.1. The Equal-Area Stratified Random Sampling Method Based on Multiple Indicator Constraints

To ensure a sufficient number of samples in complex areas and to position appropriate sampling points in homogeneous and heterogeneous areas, the equal-area stratified random sampling method was utilized, based on multiple indicators [16]. This method accomplishes two objectives: it augments the sample size for uncommon land cover categories and concurrently reduces the standard deviation in accuracy assessment [26]. The methodology primarily relies on two constraint indices, climate–population density data and landscape heterogeneity, to determine the number of validation points in each region. A hexagonal grid was constructed as the fundamental unit for assigning sampling points, in order to ensure adequate representation of each region and sufficient data for validation purposes.

First, since land cover information is intricately intertwined with climate change and human activities [29,30], climate and population density data were incorporated to generate a climate–population density layer, serving as the primary foundation for the layout of samples. Specifically, the population and un-population layers in Xinjiang were delineated based on the population density data. The climate data were reclassified into five categories (Table 3) and overlaid with the population and un-population layers to obtain the climate–population density layer.

Then, because landscape variables significantly influence the accuracy of LC products and classification results are generally more attainable in areas with homogeneous landscapes than those with heterogeneous landscapes [31], landscape heterogeneity was used as a quantitative measure of landscape fragmentation within the study area, enabling the development of a more robust land cover validation dataset. Therefore, in addition to establishing the climate–population density layer, this study also incorporated landscape heterogeneity as a criterion for the allocation of sample points. The objective was to allocate a greater number of sample points in highly heterogeneous areas, thereby establishing a more rigorous validation framework to evaluate the accuracy of the LC products [32]. The Shannon diversity index was used to describe the degree of landscape heterogeneity [33]. In a landscape ecosystem, the more diverse the land use types and the higher the fragmentation, the greater the amount of information contained in the patches, resulting in a higher calculated value of the Shannon diversity index (*SHDI*) [34]:

$$SHDI = -\sum_{i=1}^{m} (P_i)(\ln P_i) \tag{1}$$

where $P_i$ is the proportion occupied by landscape patch type *i* and *m* is the total number of land cover types in the landscape patch; the GLC_FCS30 was utilized in this study to compute the *SHDI*. The calculated *SHDI* map was classified into different layers at intervals of 0.2 for the creation of a landscape constraint layer that represents landscape heterogeneity.

Finally, in order to ensure a representative sample distribution across the Xinjiang province, the hexagonal discrete global grid system (HDGGS) was employed to gather validation datasets [6,21]. The R programming language was used to generate HDGGS grids. These grids were then employed as the foundational units for allocating samples,

while the generated climate–population density layer and the landscape heterogeneity layer were used to constrain the samples in each HDGGS grid. Specifically, all samples were first stratified and allocated to each constraint layer based on the proportional area of each constraint layer. Then, the number of HDGGS grids within each constraint layer was calculated, and samples within each constraint layer were equally distributed in each HDGGS grid.

### 3.2.2. Labeling HDLV-XJ Based on Visual Interpretation Method

Based on the aforementioned constraints, a total of 22,000 samples were allocated to the HDGGS grids based on the area of each separate region formed by the constraint layers. Since high-resolution imagery in Google Earth (version: 7.3.6.9345) is sufficient and often used to assist in the visual interpretation of datasets for validation purposes [35–37], Google Earth was used for the visual interpretation of the real land cover of each validation sample. It enables the careful examination of shapes, sizes, and patterns, thereby facilitating the assessment of human impacts on the Earth's surface [3,15,38]. A total of nine land cover types were classified, which encompassed cropland, forest, shrubland, grassland, water, permanent snow/ice, bare land, impervious, and wetland. Table 4 presents these land cover types, along with examples showcasing their appearance on Google Earth. In order to maintain the accuracy of the generated validation dataset, samples that were difficult to identify were directly excluded.

**Table 4.** The land cover types and examples of their appearance on Google Earth.

| LC Type | Typical Imagery on Google Earth |
|---------|--------------------------------|
| Cropland |  |
| Forest |  |
| Shrubland |  |
| Grassland |  |

**Table 4.** *Cont.*

| LC Type | Typical Imagery on Google Earth |
|---------|--------------------------------|
| Water |  |
| Snow/Ice |  |
| Bare land |  |
| Impervious |  |
| Wetland |  |

*3.3. Accuracy and Consistency Assessment for Land Cover Products*

3.3.1. Accuracy Assessment

The calculation of accuracy metrics, utilizing a confusion matrix, is a widely employed approach for quantitative remote-sensing accuracy assessment [31,39]. By utilizing the confusion matrix, it becomes possible to calculate the overall accuracy of the mapping results, as well as the user accuracy and producer accuracy for each land cover type. The confusion matrix offers a comprehensive evaluation of remote sensing classification outcomes, aiding in the comprehension of accuracy levels and the identification of error sources within the mapping results [40]. The calculation formulas for overall accuracy (*O.A.*), user's accuracy (*U.A.*), producer's accuracy (*P.A.*), and kappa coefficients are provided by

$$O.A. = \left( \sum_{i=1}^{m} n_{ii} \right) / n \tag{2}$$

$$U.A. = \frac{n_{ii}}{n_{+i}} \tag{3}$$

$$P.A. = \frac{n_{ii}}{n_{i+}} \tag{4}$$

$$Kappa = \frac{n \sum_{i=1}^{m} n_{ii} - \sum_{i=1}^{m} (n_{+i} + n_{i+})}{n^2 - \sum_{i=1}^{m} (n_{+i} + n_{i+})} \tag{5}$$

respectively, where $n_{ii}$ is the correctly classified pixel number of type i, m is the number of land cover types, n is the total pixel number in the study area, $n_{i+}$ is the total pixel number of type i in the validation dataset, and $n_{+i}$ is the total pixel number of type i in the ground truth data.

### 3.3.2. Consistency Analysis

The evaluation of LC product consistency can be performed considering both area and pixel levels. Area consistency examines the variations in the proportion of land cover types across different LC products. By comparing the respective areas of each land cover type among different products, it becomes possible to visually assess the consistency and diversity of these LC products [41–43]. The pixel consistency in LC products can reflect the spatial consistency of each land cover type. The pixel consistency is obtained using the spatial overlay method. The overall similarity coefficient (*OS*) and class similarity coefficient (*CS*) between different products can be calculated using [44]

$$OS = \frac{\sum_{1}^{n} XY_{ii}}{M} \times 100\% \tag{6}$$

$$CS_i = \frac{XY_{ii}}{(X_i + Y_i)/2} \times 100\% \tag{7}$$

where $X_i$ is the number of pixels of the *i*-th LC type in LC product *X*; $Y_i$ is the number of pixels of the *i*-th LC type in the comparison LC product *Y*; $XY_{ii}$ is the number of consistent pixels between *X* and *Y*; *n* is the number of land cover types, which is 9 in this case; and *M* is the total number of pixels in the study area.

## 4. Results

### *4.1. The High-Density Land Cover Validation Dataset for Xinjiang*

In order to generate a high-density and representative validation dataset for Xinjiang, the HDGGS was used to generate hexagonal grids as the basic unit for sample allocation. Figure 6a shows the results of the generated hexagonal grid units, including 1604 grids in the Xinjiang region. In addition, a total of 20,932 validation samples were generated within these grid units. The spatial distribution and distribution via the categories of these validation samples are illustrated in Figure 6b. The interpretation of land features relied on the comprehensive status observed in high-resolution imagery from Google Earth for 2020. It can be observed that the sample points are distributed across the study area. The equal-area stratified random sampling method employed in this study ensures the distribution of sample points in different areas.

Furthermore, in order to demonstrate the accuracy of the HDLV-XJ dataset that we developed, we merged the validation samples from SRS_Val and GLV_2015 to cross-compare with our HDLV-XJ product. Considering that the locations of the sample points in the three validation datasets do not correspond one-to-one, the nearest neighbor points were selected as the reference points for accuracy calculation [45]. The confusion matrix was computed using Equations (2)–(5), and the resulting values are presented in Table 5. When SRS_Val and GLV_2015 were used as reference benchmarks, the overall accuracy of our HDLV_XJ dataset exceeded 80% for both. Specifically, the producer's accuracy for bare land and snow/ice was around 90%. In the case of SRS_Val vs. HDLV-XJ, the producer's accuracy for grassland and cropland was above 90%. Although forest and shrubland showed a lower accuracy than the above land cover types, the agreement still reached 75%. Taking into account errors in the visual interpretation of the validation samples, as well as differences in the years represented by the validation datasets, the comparison results with the third-party dataset demonstrated the reliability of our HDLV-XJ dataset.

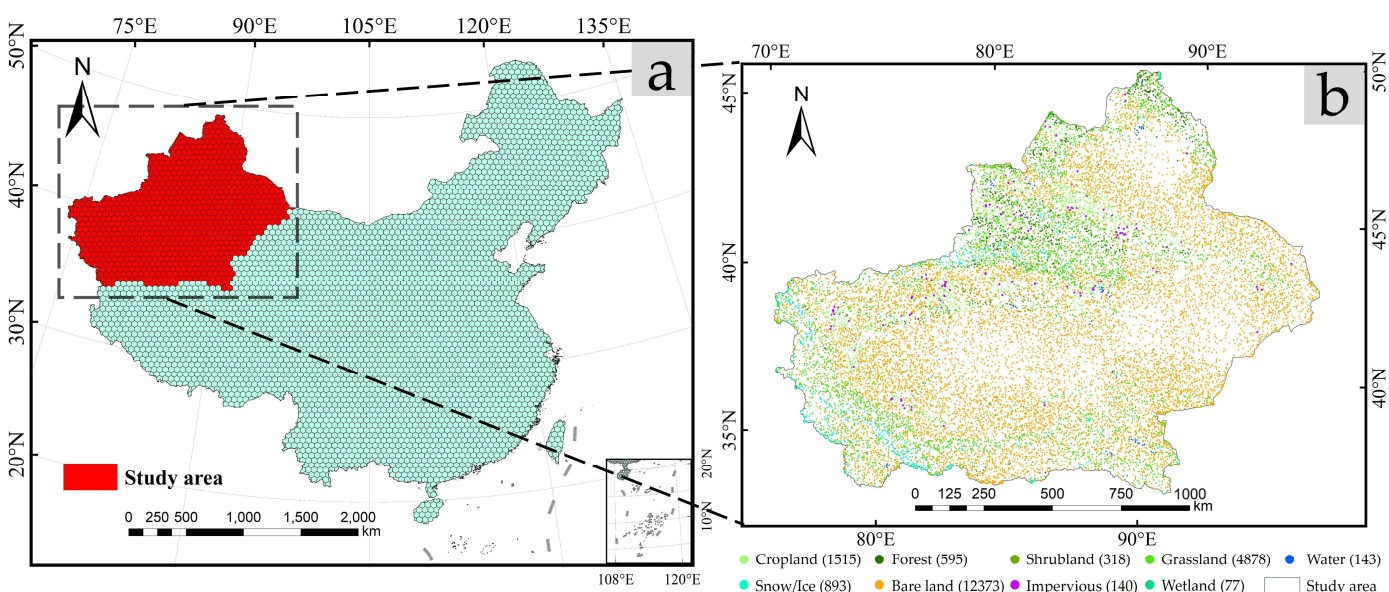

**Figure 6.** (**a**) HDGGS grids and (**b**) the spatial distribution of validation samples in HDLV-XJ.

**Table 5.** The accuracy of HDLV_XJ as compared with SRS_Val and GLV_2015.

| Classes | SRS_Val vs. HDLV-XJ | | GLV_2015 vs. HDLV-XJ | |
| --- | --- | --- | --- | --- |
| | P.A. | U.A. | P.A. | U.A. |
| Cropland | 100.00% | 77.78% | 0.00% | 0.00% |
| Forest | 75.00% | 100.00% | 0.00% | 0.00% |
| Shrubland | 75.00% | 75.00% | 0.00% | 0.00% |
| Grassland | 90.63% | 82.86% | 0.00% | 0.00% |
| Water | 0.00% | 0.00% | 0.00% | 0.00% |
| Snow/Ice | 75.00% | 50.00% | 94.44% | 77.27% |
| Bare land | 89.09% | 100.00% | 89.80% | 100.00% |
| Impervious | 0.00% | 0.00% | 0.00% | 0.00% |
| Wetland | 0.00% | 0.00% | 0.00% | 0.00% |
| O.A. | 88.68% | | 82.43% | |

*4.2. Accuracy Assessment of GLC_FCS30, GlobeLand30, and CLCD*

The confusion matrices for the three LC products in Xinjiang were calculated using the validation dataset constructed for this study, utilizing Equations (2)–(5). Based on the obtained results, the GLC_FCS30 exhibits the highest overall accuracy of 88.10%. The GlobeLand30 follows with the second highest overall accuracy of 83.58%, while the CLCD demonstrates the lowest overall accuracy of 81.57%.

The GLC_FCS30 demonstrates an overall accuracy of 88.10%, along with a kappa coefficient of 0.799 (Table 6). Regarding the producer's accuracy, forest exhibits the highest accuracy, followed by bare land, shrubland, cropland, and water. However, impervious and wetland display lower accuracies. These findings suggest that regions with homogenous surface cover types occupying larger proportions in the study area generally exhibit higher accuracy levels. Conversely, complex surface cover types often exhibit confusion with other types. For instance, wetland, which possess particularly intricate spectra, are prone to confusion with vegetation [46]. According to Table 6, approximately 49.4% of wetland validation points were incorrectly classified as vegetation types, including cropland, forest, shrubland, and grassland. In terms of user's accuracy, cropland, water, permanent snow/ice, and bare land exhibit similar accuracies to the producer's accuracy. Water demonstrates the highest user's accuracy, reaching 96.24%. This indicates the product's remarkable capability to accurately assign samples to their respective water categories.

**Table 6.** The accuracy matrix for the GLC_FCS30.

| | | Classified | | | | | | | | | |
|---|---|---|---|---|---|---|---|---|---|---|---|
| | | CRP | FST | SHR | GRS | Wat | SI | BaL | IMP | WET | Total |
| Reference | CRP | 1369 | 4 | 59 | 13 | 0 | 0 | 61 | 7 | 2 | 1515 |
| | FST | 1 | 570 | 4 | 6 | 0 | 0 | 14 | 0 | 0 | 595 |
| | SHR | 1 | 1 | 294 | 7 | 0 | 0 | 15 | 0 | 0 | 318 |
| | GRS | 130 | 204 | 51 | 3679 | 0 | 20 | 788 | 1 | 5 | 4878 |
| | Wat | 1 | 0 | 3 | 1 | 128 | 2 | 6 | 1 | 1 | 143 |
| | SI | 0 | 1 | 2 | 89 | 2 | 781 | 18 | 0 | 0 | 893 |
| | BaL | 6 | 4 | 181 | 542 | 2 | 108 | 11,513 | 1 | 16 | 12,373 |
| | IMP | 18 | 0 | 16 | 3 | 0 | 0 | 17 | 86 | 0 | 140 |
| | WET | 12 | 3 | 9 | 14 | 1 | 1 | 16 | 0 | 21 | 77 |
| | Total | 1538 | 787 | 619 | 4354 | 133 | 912 | 12,448 | 96 | 45 | 20,932 |
| | P.A. | 90.36% | 95.80% | 92.45% | 75.42% | 89.51% | 87.46% | 93.05% | 61.43% | 27.27% | |
| | U.A. | 89.01% | 72.43% | 47.50% | 84.50% | 96.24% | 85.64% | 92.49% | 89.58% | 46.67% | |
| | O.A. | | | | | 88.10% | | | | | |
| | Kappa | | | | | 0.798716 | | | | | |

Note: CRP: cropland; FST: forest; SHR: shrubland; GRS: grassland; Wat: water; SI: permanent snow/ice; BaL: bare land; IMP: impervious; WET: wetland.

The CLCD achieves an overall accuracy of 81.57%, with a kappa coefficient of 0.675 (Table 7). Considering producer's accuracy, bare land exhibits the highest accuracy at 91.01%, followed by water, cropland, and grassland. However, shrubland and wetland display relatively low producer's accuracies, possibly due to identification difficulties within the CLCD. In terms of user's accuracy, cropland and forest demonstrate higher accuracies at 93.51% and 87.84%, respectively, indicating strong classification abilities for these land cover types. Notably, the producer's accuracy and user's accuracy for the shrubland in the CLCD classifier are both zero, suggesting that no samples were correctly classified within this category in the training dataset for the CLCD classifier, as depicted in Figure 7c,d. This deficiency in training data representation for the shrubland potentially undermines the classification performance of this specific class.

**Table 7.** The accuracy matrix for the CLCD.

| | | Classified | | | | | | | | | |
|---|---|---|---|---|---|---|---|---|---|---|---|
| | | CRP | FST | SHR | GRS | Wat | SI | BaL | IMP | WET | Total |
| Reference | CRP | 1211 | 7 | 0 | 261 | 0 | 0 | 24 | 11 | 1 | 1515 |
| | FST | 3 | 289 | 0 | 295 | 0 | 1 | 4 | 3 | 0 | 595 |
| | SHR | 8 | 0 | 0 | 203 | 0 | 0 | 101 | 6 | 0 | 318 |
| | GRS | 26 | 29 | 0 | 3620 | 1 | 14 | 1173 | 14 | 1 | 4878 |
| | Wat | 0 | 1 | 0 | 11 | 115 | 1 | 11 | 4 | 0 | 143 |
| | SI | 0 | 0 | 0 | 52 | 11 | 551 | 279 | 0 | 0 | 893 |
| | BaL | 17 | 0 | 0 | 1040 | 10 | 26 | 11,261 | 19 | 0 | 12,373 |
| | IMP | 22 | 1 | 0 | 77 | 0 | 0 | 18 | 22 | 0 | 140 |
| | WET | 8 | 2 | 0 | 34 | 4 | 1 | 18 | 5 | 5 | 77 |
| | Total | 1295 | 329 | 0 | 5593 | 141 | 594 | 12,889 | 84 | 7 | 20,932 |
| | P.A. | 79.93% | 48.57% | 0.00% | 74.21% | 80.42% | 61.70% | 91.01% | 15.71% | 6.49% | |
| | U.A. | 93.51% | 87.84% | 0.00% | 64.72% | 81.56% | 92.76% | 87.37% | 26.19% | 71.43% | |
| | O.A. | | | | | 81.57% | | | | | |
| | Kappa | | | | | 0.675249 | | | | | |

Note: CRP: cropland; FST: forest; SHR: shrubland; GRS: grassland; Wat: water; SI: permanent snow/ice; BaL: bare land; IMP: impervious; WET: wetland.

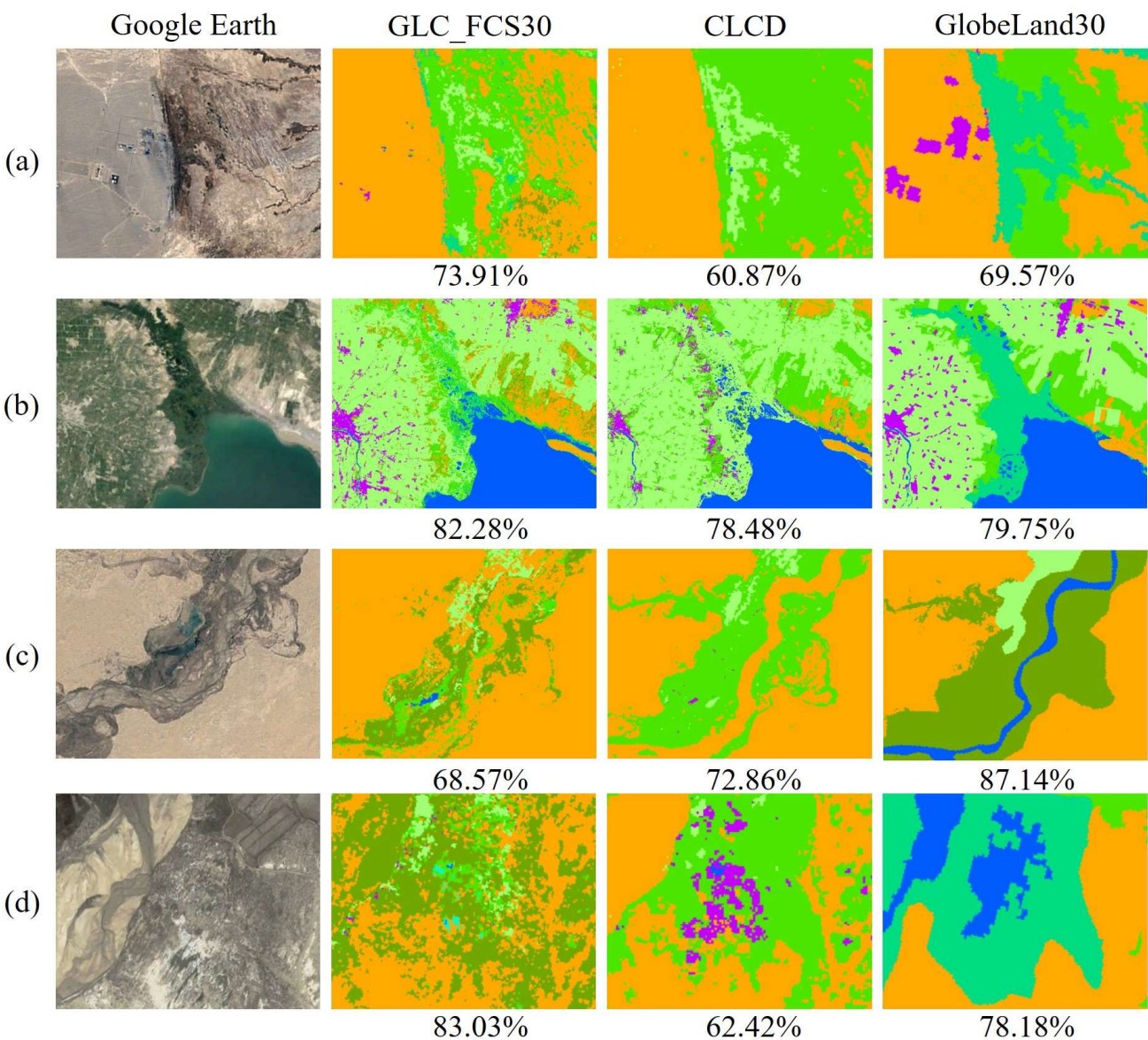

**Figure 7.** Misclassification errors of (**a**,**b**) wetland and (**c**,**d**) shrubland.

The overall accuracy of the GlobeLand30 is 83.58%, with a kappa coefficient of 0.717 (Table 8). Regarding the producer's accuracy, bare land obtains the highest accuracy at 90.28%, followed by cropland, water, and wetland. On the other hand, shrubland has a relatively low producer's accuracy of 18.24%, indicating significant challenges in classifying shrubland. As for the user's accuracy, cropland and bare land demonstrate higher accuracies of 90.13% and 89.60%, respectively, suggesting GlobeLand30's strong ability to correctly classify instances of cropland and bare land. However, the user's accuracy for shrubland is lower, at 32.77%. In summary, GlobeLand30 performs well in some land cover categories, such as cropland and bare land, but there is room for improvement in the classification of other categories.

In conclusion, it is evident that complex land cover types are more susceptible to misclassification. For example, based on Table 6, it is apparent that wetlands in GLC_FCS30 exhibit a higher confusion ratio, with over 50% of validation samples being misclassified as other types, as demonstrated in Figure 7a,b. There are also numerous instances of misclassification between similar land cover types. In particular, approximately 20% of shrubland was classified incorrectly as forest and grassland, as depicted in Figure 7c,d.

**Table 8.** The accuracy matrix for the GlobeLand30.

|  |  | Classified | | | | | | | | | |
|---|---|---|---|---|---|---|---|---|---|---|---|
|  |  | **CRP** | **FST** | **SHR** | **GRS** | **Wat** | **SI** | **BaL** | **IMP** | **WET** | **Total** |
| **Reference** | **CRP** | 1360 | 2 | 1 | 109 | 1 | 0 | 15 | 21 | 6 | 1515 |
| | **FST** | 5 | 274 | 6 | 297 | 1 | 1 | 9 | 2 | 0 | 595 |
| | **SHR** | 18 | 5 | 58 | 120 | 2 | 0 | 105 | 3 | 7 | 318 |
| | **GRS** | 48 | 58 | 46 | 3800 | 7 | 40 | 854 | 4 | 21 | 4878 |
| | **Wat** | 1 | 1 | 0 | 3 | 126 | 1 | 2 | 1 | 8 | 143 |
| | **SI** | 0 | 5 | 3 | 37 | 2 | 543 | 302 | 0 | 1 | 893 |
| | **BaL** | 59 | 9 | 61 | 988 | 25 | 33 | 11,170 | 8 | 20 | 12,373 |
| | **IMP** | 18 | 0 | 1 | 11 | 0 | 0 | 6 | 104 | 0 | 140 |
| | **WET** | 0 | 1 | 1 | 3 | 8 | 0 | 3 | 0 | 61 | 77 |
| | **Total** | 1509 | 355 | 177 | 5368 | 172 | 618 | 12,466 | 143 | 124 | 20,932 |
| | **P.A.** | 89.77% | 46.05% | 18.24% | 77.90% | 88.11% | 60.81% | 90.28% | 74.29% | 79.22% | |
| | **U.A.** | 90.13% | 77.18% | 32.77% | 70.79% | 73.26% | 87.86% | 89.60% | 72.73% | 49.19% | |
| | **O.A.** | | | | 83.58% | | | | | | |
| | **Kappa** | | | | 0.717466 | | | | | | |

Note: CRP: cropland; FST: forest; SHR: shrubland; GRS: grassland; Wat: water; SI: permanent snow/ice; BaL: bare land; IMP: impervious; WET: wetland.

### 4.3. Consistency Analysis for Global Land Cover Products

Through the spatial overlay of the three LC products, a combination of visual mapping and quantitative expression approaches was employed to illustrate the overall pixel consistency between two or more products, as well as the pixel consistency for each land cover type.

To begin, the overall similarity coefficient (*OS*) and class similarity coefficient (*CS*) were computed for different products using Equations (6) and (7), as presented in Table 9. The overall similarity coefficient between CLCD and GlobeLand30 was 83.32%, whereas between GLC_FCS30 and CLCD, it was 78.46%. Notably, the lowest overall similarity coefficient of 75.16% was observed between GLC_FCS30 and GlobeLand30. The above analysis reveals that approximately 75% of pixels share the same land cover label between any two LC products. Additionally, through overlaying the three products, an overall similarity coefficient of 69.96% was obtained, indicating a lower coefficient compared to the pairwise similarities. Consequently, it can be inferred that the land cover types are completely identical in 69.96% of the study area.

**Table 9.** Overall similarity coefficient (*OS*) (%) and individual class similarity coefficient (*CS*) (%) among different products.

| Similarity Coefficient | Three Maps | GLC_FCS30 vs. CLCD | CLCD vs. GlobeLamd30 | GLC_FCS30 vs. GlobeLand30 |
|---|---|---|---|---|
| Cropland | 66.01% | 69.76% | 80.60% | 64.47% |
| Forest | 30.95% | 42.48% | 55.20% | 35.27% |
| Shrubland | 0.00% | 0.00% | 0.00% | 2.10% |
| Grassland | 37.76% | 48.53% | 67.80% | 43.26% |
| Water | 66.13% | 71.81% | 74.80% | 61.89% |
| Snow/Ice | 51.84% | 55.52% | 72.42% | 47.02% |
| Bare land | 74.14% | 73.46% | 90.41% | 69.55% |
| Impervious | 12.14% | 17.64% | 15.44% | 40.84% |
| Wetland | 0.78% | 2.34% | 8.96% | 8.12% |
| OS | 69.96% | 78.46% | 83.32% | 75.16% |

Subsequently, the class similarity coefficients were calculated among the three products. The analysis reveals that the class similarity coefficients for bare land, water, and cropland exceed 60%, indicating a spatial correspondence of over 60% between these land cover types (Figure 8). Remarkably, the average value of the class similarity coefficient

for bare land stands is highest, at 77.14%. In contrast, wetland and shrubland exhibit the lowest class similarity coefficients, suggesting limited overlap among all three products. Additionally, the consistency for the impervious category is notably low, with an average class similarity coefficient of 21.52%.

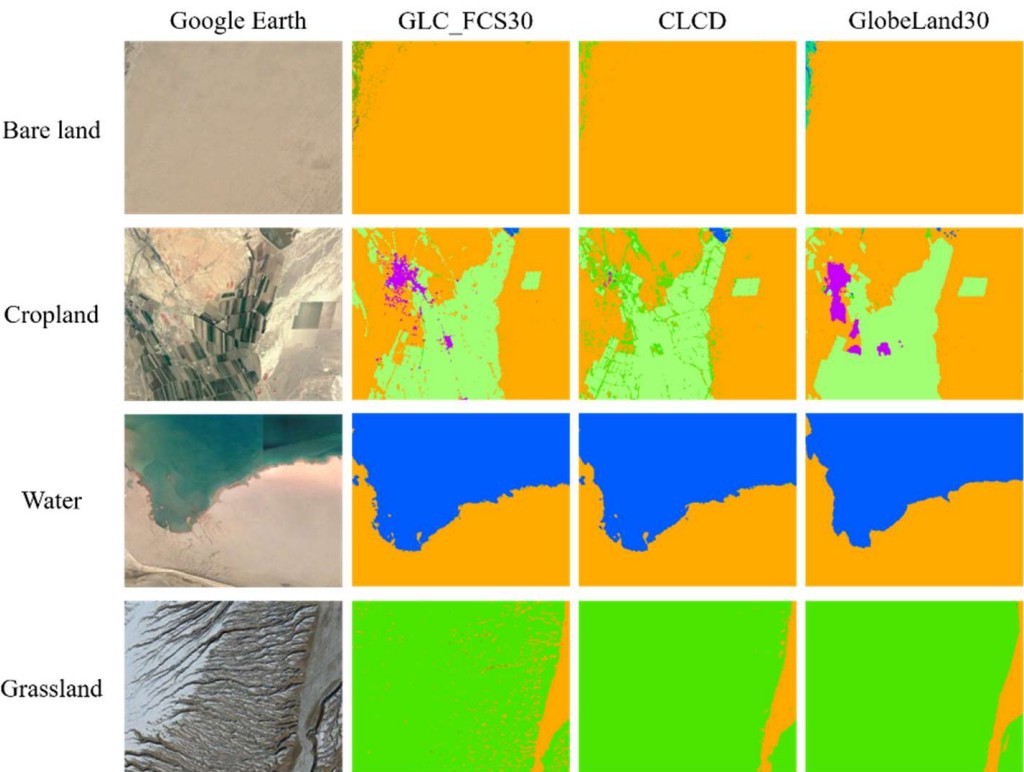

**Figure 8.** Examples of LC types with high consistency.

Simultaneously, the pixel consistency between the two products across different land cover types aligns with the aforementioned similarity of the three products. The highest consistency is observed in bare land, with class similarity coefficients of 73.46%, 90.41%, and 69.55% for GLC_FCS30 vs. CLCD, CLCD vs. GlobeLand30, and GLC_FCS30 vs. GlobeLand30, respectively. Following that, cropland exhibits the next highest consistency, with class similarity coefficients ranging from 64% to 80%. Conversely, the class similarity coefficients for shrubland and wetland are below 9%.

The average areas of the nine land classes for the three LC products, as presented in Table 10, can be sorted in descending order based on their average area: bare land, grassland, cropland, permanent snow/ice, forest, shrubland, water, impervious, and wetland. Notably, the average areas of bare land, grassland, and cropland surpass 90,000 km$^2$. Their respective average areas are 1,102,577.54 km$^2$; 350,050.94 km$^2$; and 98,989.83 km$^2$. Bare land covers 66.22% of the region, making it the predominant geographical landscape.

Table 11 presents the comparative area results for each land class in three LC products: GLC_FCS30, CLCD, and GlobeLand30. There is a relatively high consistency in the area measurements for bare land, cropland, and permanent snow/ice across the three products. The consistency is moderate for water and impervious. However, significant variations exist in the area measurements for some land classes, particularly for shrubland and wetland, indicating low consistency. The area of shrubland in GLC_FCS30 is nearly four times larger than in GlobeLand30, while CLCD only reports shrubland of 1.27 km$^2$, significantly smaller than both GLC_FCS30 and GlobeLand30. Moreover, GlobeLand30 indicates a larger wetland area of 9197.70 km$^2$ compared to the other two products. In contrast, CLCD reports a mere 500.83 km$^2$ of wetland area, accounting for a mere 0.03% of the total area.

**Table 10.** The average area of LC types of the LC products (unit: km$^2$).

| LC Type | Mean Area | Percentage Mean |
|---|---|---|
| Cropland | 98,989.83 | 5.95% |
| Forest | 29,424.22 | 1.77% |
| Shrubland | 18,850.94 | 1.13% |
| Grassland | 350,050.94 | 21.03% |
| Water | 11,369.57 | 0.68% |
| Snow/Ice | 42,092.11 | 2.53% |
| Bare land | 1,102,577.54 | 66.22% |
| Impervious | 7339.36 | 0.44% |
| Wetland | 4202.49 | 0.25% |

**Table 11.** The areas of LC types of the LC products (unit: km$^2$).

| LC Type | GLC_FCS30 | Percentage in GLC_FCS30 | CLCD | Percentage in CLCD | GlobeLand30 | Percentage in GlobeLand30 |
|---|---|---|---|---|---|---|
| Cropland | 104,931.34 | 6.30% | 87,744.63 | 5.27% | 104,293.52 | 6.26% |
| Forest | 48,311.59 | 2.90% | 18,576.16 | 1.12% | 21,384.91 | 1.28% |
| Shrubland | 44,592.01 | 2.68% | 1.27 | 0.00% | 11,959.54 | 0.72% |
| Grassland | 307,358.94 | 18.46% | 383,017.16 | 23.01% | 359,776.72 | 21.61% |
| Water | 10,028.85 | 0.60% | 10,792.64 | 0.65% | 13,287.22 | 0.80% |
| Snow/Ice | 55,086.92 | 3.31% | 35,631.59 | 2.14% | 35,557.83 | 2.14% |
| Bare land | 1,084,205.57 | 65.12% | 1,123,584.54 | 67.49% | 1,099,942.51 | 66.07% |
| Impervious | 7472.84 | 0.45% | 5048.18 | 0.30% | 9497.05 | 0.57% |
| Wetland | 2908.94 | 0.17% | 500.84 | 0.03% | 9197.70 | 0.55% |

An analysis of area consistency between GLC_FCS30 and GlobeLand30 exposes notable variations across different land cover types. The areas of bare land in both products exhibit high similarity, encompassing approximately 66% of the total study area. A similar pattern emerges in cropland, where GLC_FCS30 reports an area of 104,931.34 km$^2$ and GlobeLand30 reports an area of 104,293.52 km$^2$. However, for other categories, the consistency between the two products is relatively low.

## 5. Discussion

### 5.1. The Advantages of the HDLV-XJ Dataset

In this study, we combined multiple indicators and the equal-area stratified sampling method to generate over 20,000 validation samples in Xinjiang province. The high-density HDLV-XJ validation dataset ensured a sufficient quantity of validation data in both homogeneous and heterogeneous regions. In order to demonstrate the advantages of our dataset, the *SHDI* of Xinjiang was calculated and categorized into 10 levels to exhibit the degree of landscape fragmentation in this region (Figure 9a). A higher level indicates a greater diversity of land-use types and a higher degree of fragmentation. Figure 9b illustrates the sample proportions at different landscape heterogeneities. In addition, we also assigned the same number of samples into this region using random sampling as a comparison. As illustrated in Figure 9b, the sample proportion of HDLV-XJ is higher than the distribution proportion of random sampling results in the heterogeneous regions. The random sampling allocation results in an excessive number of samples in homogeneous regions, resulting in a lower representation in heterogeneous regions. As a comparison, our HDLV-XJ dataset takes into account the distribution of samples in heterogeneous regions while ensuring that homogeneous regions are allocated with sufficient samples.

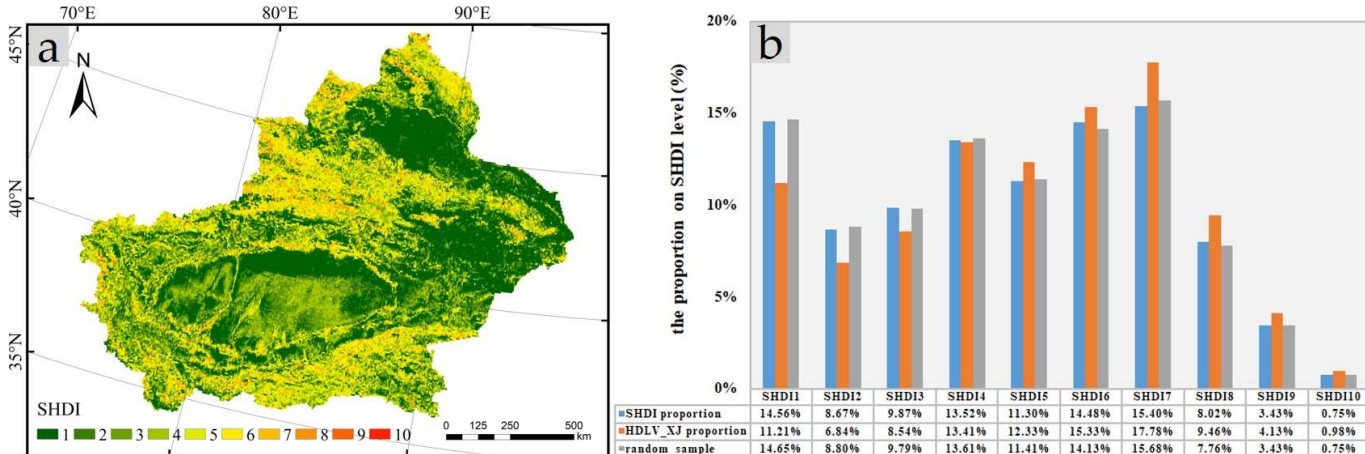

**Figure 9.** (**a**) The Shannon diversity index (*SHDI*) and (**b**) sample proportions of different landscape heterogeneity levels. The calculated *SHDI* map was classified into different layers at intervals of 0.2.

In addition, by comparing the validation point quantities of two global land cover validation datasets (SRS_Val, GLV_2015) and our HDLV-XJ dataset across different land cover types (Table 12), our HDLV-XJ dataset showed considerably higher sample numbers for each land cover type compared to the other datasets. This indicates that the global-scale validation data cannot accommodate regional-scale accuracy assessments. Our HDLV-XJ provides representative validation data with sufficient samples for rare categories (such as water (143 HDLV-XJ samples vs. 3 SRS_Val samples and 1 GLV_2015 samples) and impervious (140 HDLV-XJ samples vs. 5 SRS_Val samples and 2 GLV_2015 samples), enabling a more accurate assessment of the accuracy of land cover products in the Xinjiang area. Our HDLV-XJ product provides scientific data to support the use of land cover products in the Xinjiang region.

**Table 12.** The comparison of the validation point quantities for each LC type among the three datasets.

| LC Type | CRP | FST | SHR | GRS | Wat | SI | BaL | IMP | WET | Total |
|---------|-----|-----|-----|-----|-----|-----|-----|-----|-----|-------|
| HDLV-XJ | 1515 | 595 | 318 | 4878 | 143 | 893 | 12,373 | 140 | 77 | 20,932 |
| SRS_Val | 55 | 23 | 19 | 143 | 3 | 28 | 381 | 5 | 2 | 659 |
| GLV_2015 | 0 | 0 | 32 | 0 | 1 | 96 | 283 | 2 | 0 | 403 |

Note: CRP: cropland; FST: forest; SHR: shrubland; GRS: grassland; Wat: water; SI: permanent snow/ice; BaL: bare land; IMP: impervious; WET: wetland.

### 5.2. Analysis of the Relationship between Different Environment Conditions and the Performance of Land Cover Products

Due to the potential impact of landscape heterogeneity on LC mapping accuracy [26,47–49], we conducted an analysis to examine the relationship between landscape heterogeneity and the accuracy of LC products (Figure 10). The results showed that the accuracy of all three LC products significantly decreased (*p*-value: <0.05) with increasing landscape heterogeneity. Specifically, the LC product with the highest slope of decrease was CLCD (with a slope of −0.32), followed by GLC_FCS30 (with a slope of −0.22), and finally GlobeLand30 (with a slope of −0.17). Furthermore, the accuracy of all three LC products can generally reach 80% in homogeneous areas (low heterogeneity). In particular, the accuracy of GLC_FCS30 is close to 0.9 when the heterogeneity is below 0.1. This suggests that the degree of landscape heterogeneity indeed has a significant negative impact on LC mapping accuracy, and the results of the three products are more reliable in homogeneous areas. Additionally, GlobeLand30 generally exhibits the most robust performance in areas with high landscape heterogeneity. As can also be observed from Figure 10d–k, the consistency of classification results for the three LC products varied significantly in areas with high landscape heterogeneity, indicating a greater

degree of uncertainty. The accuracy in these regions ranges from 60% to 80%. Therefore, in order to enhance the accuracy of LC products in complex regions, producers may need to pay more attention to transitional areas where landscape heterogeneity is high. These areas exhibit higher probabilities of misclassification compared to homogeneous regions.

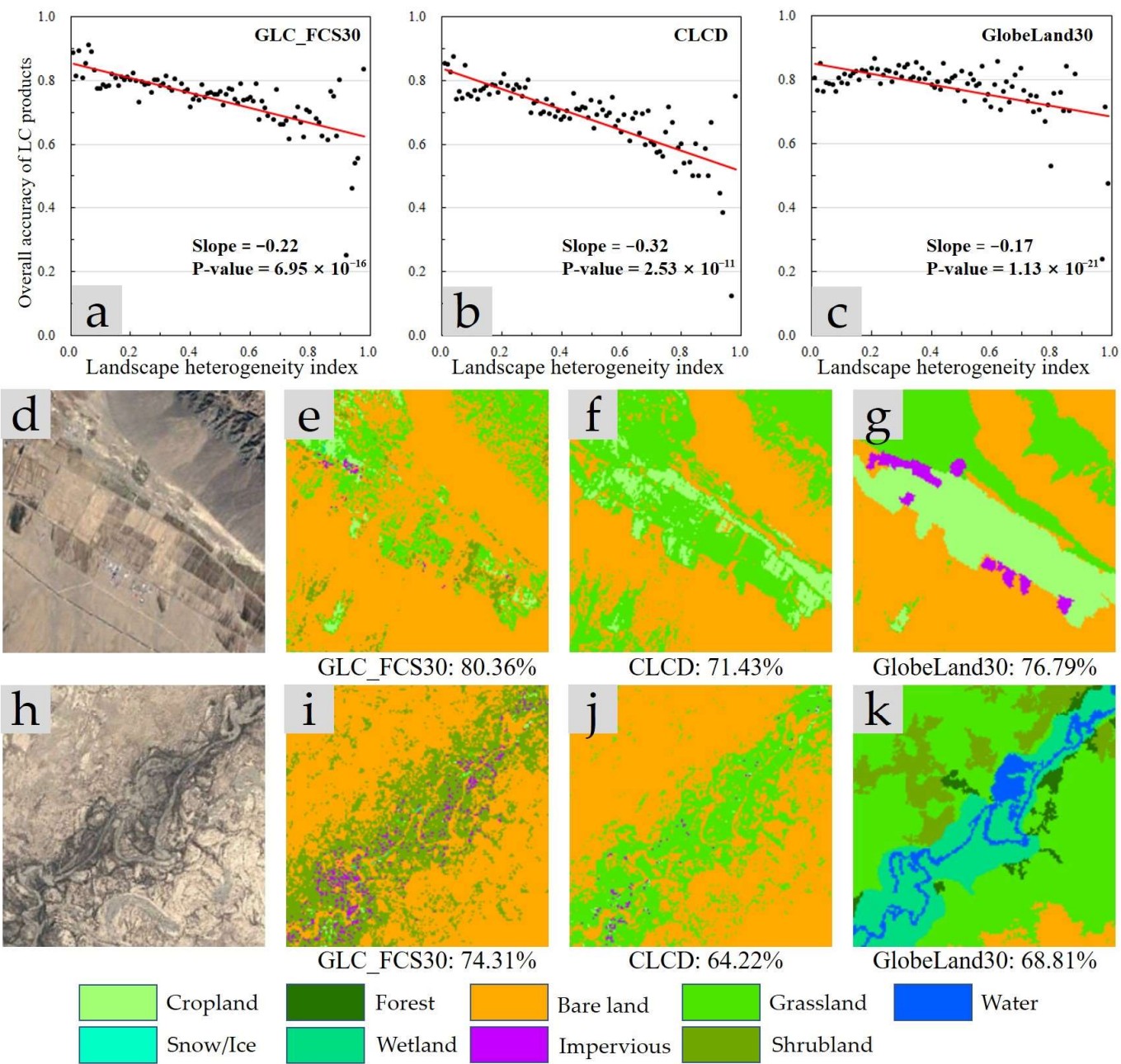

**Figure 10.** (**a**–**c**) The relationship between landscape heterogeneity and overall accuracy for LC products and (**d**–**k**) the comparison of accuracy among three products in heterogeneous regions.

Then, we analyzed the relationship between the area of land cover types and the mapping classification accuracy (Figure 11). It can be seen that the accuracy of most LC types in GLC_FCS30 is relatively stable, particularly performing well in cases with small areas. However, the classification accuracy for impervious surfaces and wetlands is still limited. Due to their strong spectral heterogeneity, impervious surfaces and wetlands are also recognized as land cover classes that are relatively difficult to classify accurately [3]. As a comparison, apart from water bodies, the mapping accuracy of each land cover class in CLCD decreases noticeably as the area decreases, with a correlation coefficient (r) of 0.56

between area and accuracy. Due to the significant differences in spectral characteristics between water bodies and other land cover classes [50], water bodies are relatively easy to identify and can maintain a high level of accuracy in CLCD. Furthermore, GlobeLand30 exhibits a similar pattern to CLCD, but its accuracy for impervious surfaces and wetlands, which are small-area land classes, is still higher than that of GLC_FCS and CLCD. This may be attributed to the incorporation of manual post-processing after classification. Therefore, GLC_FCS30 performs the best in handling small-area land classes, but its classification ability for impervious and wetland still needs improvement. On the other hand, GlobeLand30 demonstrates good classification performance in impervious and wetland classes.

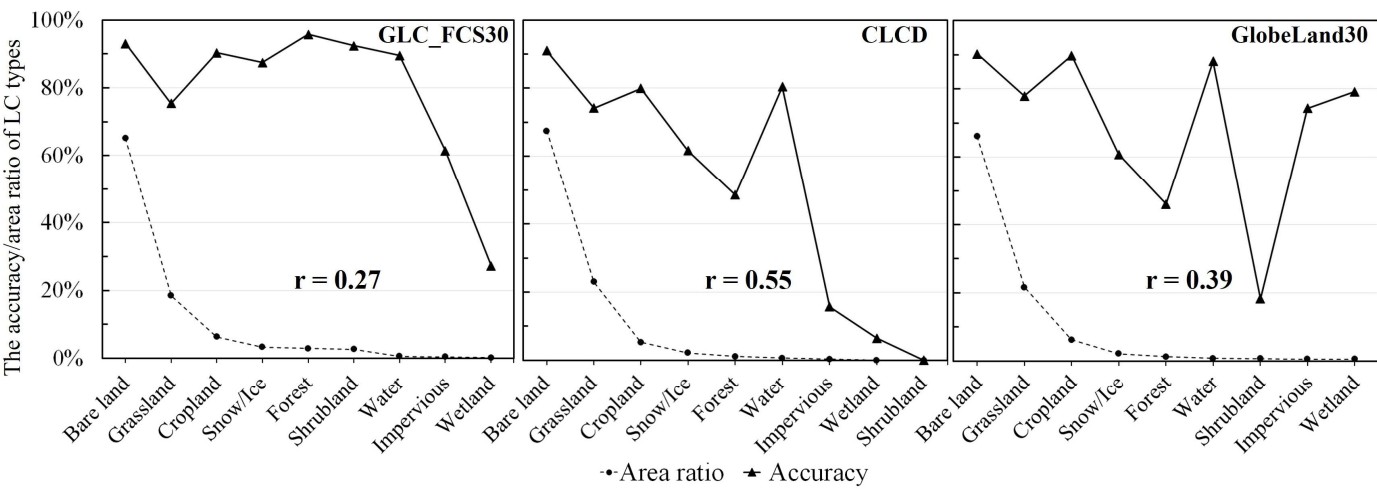

**Figure 11.** The relationship between area ratio and accuracy of LC types.

Based on the above, the performance of CLCD is more sensitive to landscape fragmentation compared to GlobeLand30 and GLC_FCS30 (Figure 10). The mean landscape heterogeneity index of Xinjiang is approximately 0.65, which is relatively high; this may be the reason for the relatively lower accuracy of CLCD compared to the other two LC products. Furthermore, it can be seen that GlobeLand30 and CLCD are more sensitive to the area ratio of land classes compared to GLC_FCS30 (Figure 11). Because our HDLV-XJ is a regionally intensive validation dataset with a high density of samples, there are still a significant number of samples (as shown in Table 12), even in land classes with relatively low coverage (such as cropland, snow and ice, forest, and shrubland) in Xinjiang. Therefore, this leads to lower assessment accuracies of CLCD and GlobeLand30 for these land classes, which in turn results in the highest validation accuracy for GLC_FCS30 in Xinjiang. The differences in mapping methods may be the main reason for the variations in accuracy among the three LC products. From Table 13, it can be seen that CLCD and GlobeLand30 are both based on global classification models, while GLC_FCS30 is based on a local adaptive classification model. Due to the requirement of ensuring overall accuracy across a large-scale area during training, there may be significant differences in the number of samples available in local areas when building global classification models [51,52].In contrast, the local adaptive classification strategy divides the large-scale area into different sub-regions and constructs training data within each sub-region [1]. Therefore, this mapping strategy can better balance the number of training data for each land cover in local areas [1]. However, GLC_FCS30 still faces challenges in accurately extracting land cover types such as wetlands and impervious surfaces. On the other hand, GlobeLand30 incorporates manual post-processing in its mapping process, which further improves the accuracy of these fragmented land cover types [3]. As a result, GlobeLand30 has a higher level of accuracy in these cases. Therefore, when conducting large-scale LC mapping, the local adaptive classification strategy may be more appropriate. Additionally, producers should also pay attention to individually addressing land cover types that are difficult to accurately identify, such as wetlands and impervious surfaces. This ensures that these

specific land cover types receive appropriate handling and improves the overall accuracy of the mapping process.

**Table 13.** The mapping methods of 30-m GLC_FCS30, GlobeLand30, and CLCD.

| LC Product | Method | Literature |
|---|---|---|
| GLC_FCS30 | Local adaptive random forest models were trained for each $5° \times 5'$ geographical grid element to generate the land-cover maps. | Zhang et al. (2021) [1] |
| CLCD | A global random forest classifier was trained to classify the whole of China. | Yang et al. (2021) [5] |
| GlobeLand30 | A global pixel- and object-based classification model was applied to classify global land covers, and a knowledge-based interactive post-process was applied to improve the mapping accuracy. | Chen et al. (2015) [3] |

*5.3. Limitations*

There are still some limitations in the present research, mainly in three aspects: (1) The constructed HDLV-XJ dataset introduced climatic factors, population density, and a hexagonal grid combined with landscape heterogeneity to design the sampling scheme, ensuring a representative and effective sample layout. However, this dataset is only applicable for land cover validation for Xinjiang in 2020, which hinders its use in long-term and large-scale accuracy assessment studies. To address this limitation, the study will explore extending the validation dataset through the combination of time-series remote sensing images and fitting algorithms and encourage public participation in the construction and interpretation of the validation dataset through open collaboration. (2) The generated validation dataset can only validate the nine primary land cover types and is not suitable for validating the secondary classification system. To overcome this limitation, the study will incorporate multiple sources of geospatial data and design a more detailed classification system. This will allow for the interpretation and validation of more detailed land cover categories. (3) Considering the importance of 30-m resolution land cover products in fine-resolution long-term monitoring, we only validated three mainstream 30-m land cover products in this study. However, with the recent availability and sharing of 10-m-resolution Sentinel data, 10-m land cover products are also gradually being generated. In the future, we will also include 10-m land cover products for further evaluation.

**6. Conclusions**

Fine-resolution LC products have been developed in recent years. However, the accuracy evaluation of the developed LC products is typically conducted at the global and national levels, with limited consideration for their accuracy and applicability in regional areas. Therefore, conducting a comprehensive accuracy assessment and consistency analysis of LC products in local areas is crucial for users to effectively compare the performance of different products. This study examined Xinjiang as the research area and constructed a high-density land cover validation dataset (HDLV-XJ, containing 20,932 validation samples) based on multi-source remote sensing data for Xinjiang. The accuracy and consistency of three mainstream land cover products in 2020 with a resolution of 30-m (GLC_FCS30, CLCD, GlobeLand30) were analyzed based on the constructed validation dataset.

The results indicated that the CLC_FCS30 exhibited the highest overall accuracy (88.10%) in Xinjiang, followed by GlobeLand30 (with an overall accuracy of 83.58%). By contrast, CLCD demonstrated the lowest overall accuracy of 81.57%. In terms of consistency, Xinjiang demonstrates high-consistency patterns for bare land, farmland, and water. Specifically, the average consistencies between the GLC_FCS30 and GlobeLand30, GLC_FCS30 and CLCD, and GlobeLand30 and CLCD are 77.14%, 70.21%, and 68.66%, respectively. However, this consistency drops significantly when it comes to wetland and shrubland, with a value of less than 1% for each. Furthermore, among the LC products, GlobeLand30 demonstrated the highest performance in regions characterized by high landscape fragmentation. On the other hand, GLC_FCS30 emerged as the superior product in areas with uneven proportions of land cover types. Additionally, the utilization of a local

adaptive classification mapping strategy offers significant advantages in enhancing the accuracy of land cover mapping. The study provided important insights for the application of current land cover products in Xinjiang, uncovering both the accuracy and limitations associated with these products.

**Author Contributions:** Conceptualization, X.C. and J.L.; funding acquisition, X.C.; methodology, X.C., J.L. and Y.R.; software, J.L. and Y.R.; visualization, J.L. and Y.R.; writing—original draft preparation, J.L. and X.C.; writing—review and editing, J.L., Y.R. and X.C. All authors have read and agreed to the published version of the manuscript.

**Funding:** This research was funded by the National Natural Science Foundation of China (grant no. 42301336) and the National Key Research and Development Program of China (grant no. 2021YFE0117800).

**Data Availability Statement:** Our code and developed high-density land cover validation dataset for Xinjiang (HDLV-XJ) in 2020 are now available at https://doi.org/10.5281/zenodo.10029022.

**Acknowledgments:** The authors sincerely thank the production agencies that provided free land cover datasets and validation reference datasets.

**Conflicts of Interest:** The authors declare no conflicts of interest.

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
