# Peer review of "Regional Accuracy Assessment of 30-Meter GLC_FCS30, GlobeLand30, and CLCD Products: A Case Study in Xinjiang Area"

_remotesensing, doi:10.3390/rs16010082_

Round 1
Reviewer 1 Report
Comments and Suggestions for Authors
Overall a very well written and useful paper. With the increasing number of large-scale land cover products it is difficult for users to understand the relative stenghts and weaknesses of products. Your methods will be appropriate for other maps and other locations.
I have a few minor comments. Table 1 and Table 3 duplicate information. I don't think they both need including, as it is clear the CLC (ID) is used as the 'harmonise' class. You just need table 1.
You mention that GlobLand30 has ten classes, but the table only shows nine (there no class 70).
Your accuracy/correspondence matrices would be clearer if they indicated what the columns and rows represent, ie., which are the classified data, which are the reference data.
Good paper. I recommend for publication.
Reviewer 2 Report
Comments and Suggestions for Authors
This paper evaluated several LULC products in a regional scale. There shows some problems as follow.
Insufficient originality and contribution to research: Although this study assessed the accuracy and consistency of existing land cover products in Xinjiang, it fails to demonstrate significant originality or notable contributions to existing research. The research methods and analyses appear to follow standard remote sensing data evaluation procedures, lacking innovative analysis methods or novel theoretical perspectives.
Limitations of localized accuracy validation study for large-scale data: Due to the focus on Xinjiang, this study may not directly assess the applicability of data products in other large-scale regions, which is a limitation of accuracy validation research. This is because land classification products are typically designed to cover larger areas, so there may be certain limitations in localized regions.
Inadequate interpretation and discussion of results: While the paper provides accuracy and consistency data for three land cover products, it falls short in explaining the reasons behind these results.
Lack of guidance for future research directions: The paper fails to propose clear suggestions or potential research directions for future studies, particularly regarding improving the accuracy and consistency of land cover products in similar complex areas.
However, if the authors can make significant improvements and additions in the aforementioned aspects, this research could potentially provide valuable insights into regional applications and assessments of land cover product.
Reviewer 3 Report
Comments and Suggestions for Authors
The manuscript titled Regional Accuracy Assessment of 30-meters Comparison and Assessment of Different Land Cover Datasets on the Cropland in Northeast China. Remote Sensing, A Case Study in Xinjiang area after major revision
The abstract
This section could be expanded a little. In this section, the authors only need to summarize the most important results of this research in two sentences. In this section it is also necessary to say more about the main methodology.
Keywords
The authors need to add one more keyword
Introduction
This section needs to be expanded to include the sentences that better explain the methods and procedures of the 30-meter images. The main problem for me is how the authors analyze the exact (local or regional) data with this kind of resolution.
In this part of the manuscript, the authors need to better indicate step by step what GIS or remote sensing methods were used in this study.
There are many good references that can be read and cited in this research.
I strongly recommend the authors to cite and analyze two valuable references with very similar GIS methods and approaches.
These references are:
- Valjarević, A., Popovici, C., Štilić, A. et al. Cloudiness and water from cloud seeding in connection with plants distribution in the Republic of Moldova. Appl Water Sci 12, 262 (2022). https://doi.org/10.1007/s13201-022-01784-3.
- Cui, P., Chen, T., Li, Y., Liu, K., Zhang, D., & Song, C. (2023). Comparison and Assessment of Different Land Cover Datasets on the Cropland in Northeast China. Remote Sensing, 15(21), 5134. https://doi.org/10.3390/rs15215134.
Section Study Area and Data
The authors need to add more about the geographical location and especially more about the climate and weather patterns in the area.
Figure 1, the legend below the figure needs to be explained.
Table 1, the source of the table must be given or the authors must explain how they created this table.
Figure 2, how did the authors create this map? Did the author use the finished vector file or only raster? Perhaps an unclassified or classified image classification was used?
Lines 151-153: If these climate classifications belong to the Koppen or mixed Koppen classification or any other climate classification, this should be explained in more detail.
Figure 3: Why did the authors show the Koopen-Geiger climate classification for the whole earth, what does the climate classification map for the Xinjiang area look like? Please explain better.
Figure 4,
The authors need to better explain what the spatial distribution of Figure 4 is, it is really important.
The classification of forest areas is usually by deciduous, mixed and broadleaf forests. How the authors divided the forest into several of the three classes, they should explain better.
Subsection Labeling HDLV-XJ based on the visual interpretation method
How did the authors statistically analyze the more than 22,000 samples by using machine learning approaches?
Figure 6, how is the validation procedure approved when using each training network?
Figrure 7, what percentage of accuracy, as the authors shared country characteristics, they are analyzed pixel methods?
In my opinion, for this rank of the journal of the discussion section is mandatory!!!
In this section, the methods can be divided into, for example
-numerical
-statistical
-Geographical Information Systems (GIS)
-Remote sensing,
Conclusion
This section could be expanded with a few sentences.
Overall, the conclusion is too short for this rank of journal
The paper has the scientific potential to be published after revision
I recommend a major revision
Good luck to the authors
Reviewer#2
Round 2
Reviewer 2 Report
Comments and Suggestions for Authors
Thank you for your serious and meticulous reply. I was impressed by your serious attitude in responding. Formally speaking, you have responded to each comment I have made. But in terms of content, I believe you did not accurately answer the questions about lack of innovation and contribution, as well as the analysis of the underlying reasons for the conclusions you have drawn. The paper compares the accuracy of three 30 meter data products within a regional area, Xinjiang. Firstly, other publicly available classified products have emerged, including 10 meter resolution; Secondly, at the methodological level, I did not find significant innovation. What I hope to see is an explanation of the contribution and innovation of your work to remote sensing field or related field, as well as the factors that led to the conclusions you have discovered, rather than only using your own results to illustrate your own conclusions. It would be great if there could be an analysis and explanation of the underlying mechanism. For example, why does product A perform better in a certain region than others. What inspiration or impact does your conclusion have on others.
Reviewer 3 Report
Comments and Suggestions for Authors
In my opinion, the manuscript entitled Regional Accuracy Assessment of 30-meters GLC_FCS30, GlobeLand30, and CLCD Products: A Case Study in Xinjiang area can be accepted for the publication
The authors have responded to all my suggestions and recommendations. They have corrected all misprints and errors in the text.
Sincerely,
Reviewer#2
Author Response
请参阅附件。
